# Diosmetin Targeted at Peroxisome Proliferator-Activated Receptor Gamma Alleviates Advanced Glycation End Products Induced Neuronal Injury

**DOI:** 10.3390/nu14112248

**Published:** 2022-05-27

**Authors:** Mei Chou Lai, Wayne Young Liu, Shorong-Shii Liou, I-Min Liu

**Affiliations:** 1Department of Pharmacy and Master Program, Collage of Pharmacy and Health Care, Tajen University, Pingtung 90741, Taiwan; meei@tajen.edu.tw (M.C.L.); ssliou@tajen.edu.tw (S.-S.L.); 2Department of Urology, Jen-Ai Hospital, Taichung 41265, Taiwan; waynedoctor@gmail.com; 3Center for Basic Medical Science, Collage of Health Science, Central Taiwan University of Science and Technology, Taichung 406053, Taiwan

**Keywords:** diosmetin, AGEs, Alzheimer’s disease, SH-SY5Y cells, endoplasmic reticulum stress, PPAR*γ*

## Abstract

The present study aimed to evaluate the role of diosmetin in alleviating advanced glycation end products (AGEs)-induced Alzheimer’s disease (AD)-like pathology and to clarify the action mechanisms. Before stimulation with AGEs (200 μg/mL), SH-SY5Y cells were treated with diosmetin (10 μmol/L), increasing cell viability. The induction of AGEs on the reactive oxygen species overproduction and downregulation of antioxidant enzyme activities, including superoxide dismutase, glutathione peroxidase, and catalase, were ameliorated by diosmetin. Amyloid precursor protein upregulation, accompanied by increased production of amyloid-β, caused by AGEs, was reversed by diosmetin. In the presence of diosmetin, not only β-site amyloid precursor protein cleaving enzyme1 expression was lowered, but the protein levels of insulin-degrading enzyme and neprilysin were elevated. Diosmetin protects SH-SY5Y cells from endoplasmic reticulum (ER) stress response to AGEs by suppressing ER stress-induced glucose regulated protein 78, thereby downregulating protein kinase R-like endoplasmic reticulum kinase, eukaryotic initiation factor 2 α, activating transcription factor 4, and C/EBP homologous protein. Diosmetin-pretreated cells had a lower degree of apoptotic DNA fragmentation; this effect may be associated with B-cell lymphoma (Bcl) 2 protein upregulation, Bcl-2-associated X protein downregulation, and decreased activities of caspase-12/-9/-3. The reversion of diosmetin on the AGEs-induced harmful effects was similar to that produced by pioglitazone. The peroxisome proliferator-activated receptor (PPAR)*γ* antagonist T0070907 (5 μmol/L) abolished the beneficial effects of diosmetin on AGEs-treated SH-SY5Y cells, indicating the involvement of PPAR*γ*. We conclude that diosmetin protects neuroblastoma cells against AGEs-induced ER injury via multiple mechanisms and may be a potential option for AD.

## 1. Introduction

Alzheimer’s disease (AD) is one of the progressive neurodegenerative diseases, and memory loss and cognitive impairment is the most apparent sign [1]. Amyloid-β (Aβ) peptides misfolding and aggregation is recognized as the leading cause of the toxic Aβ oligomers formation and the Aβ plaques deposition in the brain, representing the hallmarks of AD [2]. Neither cure nor treatment to alter the disease progression of AD has been observed until now [3]. Acetylcholinesterase inhibitors are available to help alleviate the symptoms, but these drugs’ efficacy is limited due to the adverse effects [4]. Aggravated achievement in effective therapeutic and preventive interventions is necessary [3].

Diabetes has been considered a decisive risk factor for AD development [5]. The pathology associated with diabetes-mediated AD involves chronic hyperglycemia accelerating the reaction between glucose and proteins to lead to advanced glycation end products (AGEs) [6]. AGEs interact with the cell surface receptors to trigger reactive oxygen species (ROS) production, following downstream stimulation of the amyloid precursor protein (APP) processing pathways, resulting in the Aβ and related molecules secretion [6]. Not only are Aβ generation from amyloidogenic processing of the APP, impairments in Aβ clearance or enzyme-mediated Aβ degradation considered to cause the accumulation and aggregation of Aβ [7]. The aggregated Aβ interferes with several cellular processes and results in the endoplasmic reticulum (ER) stress triggering a complex multistep cascade, leading to neuronal death and dementia [8]. Due to shared pathologies between AD and diabetes, the drugs used to treat diabetes may be a suitable therapeutic option for AD [9].

Peroxisome proliferator-activated receptors (PPARs) are ligand-dependent transcription factors; drugs that activate the PPAR*γ* subtype, increase insulin sensitization, and modulate glucose and lipid metabolism, commonly used to treat diabetes [10]. Pioglitazone belongs to thiazolidinedione and is a highly selective PPAR*γ* agonist found to reverse memory impairment and biochemical changes in a mouse model of type 2 diabetes mellitus and AD [11]. A clinical study has reported that pioglitazone improves cognition in patients with mild to moderate Alzheimer’s disease and type 2 diabetes mellitus [12]. Furthermore, meta-analysis studies indicate that pioglitazone treatment may offer therapeutic benefits in early or mild-to-moderate AD [13]. Some of the most common adverse effects of conventional PPAR agonists include edema and heart enlargement limited to clinical use [14].

Citrus plants belong to the economic crops and are consumed worldwide; they are also one of the primary sources of human health-promoting nutrients [15]. Diosmin (3′,5,7-trihydroxy-4′-methoxyflavone-7-rutinoside) is one flavonoid from citrus, which is composed of nutraceutical products for oral treatment with chronic venous insufficiency and varicose veins [16]. An extensive study has indicated that diosmin possesses diverse pharmacological activities, including anti-inflammatory, antihyperlipidemic, antihyperglycemic, antioxidant, anti-cancer, and free radical scavenging effects [16]. Diosmin lowering cerebral Aβ levels to alleviate cognitive impairment in AD mouse models has been documented; this compound seems to be a potential candidate for AD therapy [17]. Unfortunately, the exact mechanism of diosmin effectiveness in ameliorating AGEs-mediated AD pathology’s aggravation remains unknown.

Diosmin, like most flavonoids, is poorly soluble, leading to a low dissolution rate and impaired uptake from the gastrointestinal tract [18]. Due to the solubility of diosmin under high pH being poor, the biological effects of oral diosmin may be mediated by its aglycone, diosmetin (3′,5,7-trihydroxy-4′-methoxyflavone) [19]. Following oral administration, diosmin is quickly hydrolyzed by enzymes from the intestinal microflora into diosmetin, then absorbed. Because of its poor solubility and low bioavailability, diosmin thus requires high oral dosages and must be administered for extended periods [16]. In addition, a small amount of unconjugated form of diosmetin was detected in the plasma of healthy volunteers receiving oral administration of diosmetin [20]. When evaluating the efficacy of diosmin, considering the activity of diosmetin is thus necessary. According to an in vitro experiment, diosmetin has been more effective than diosmin [21]. We, therefore, evaluated the bioactivities of diosmetin in an in vitro cellular model. Human neuroblastoma SH-SY5Y cells can be differentiated from a neuroblast-like state into mature human neurons, induced by AGEs as an in vitro model of AD-like pathology [22]. In the present study, pioglitazone was used as a positive control to clarify whether diosmetin had protective effects on AGEs-induced Aβ neurotoxicity in SH-SY5Y cells and explore the possible mechanisms.

## 2. Materials and Methods

### 2.1. Cell Lines and Culture Conditions

Human SH-SY5Y neuroblastoma cells (no. CRL-266) were purchased from American Type Culture Collection (Manassas, VA, USA) and cultured in DMEM/F12 medium (Sigma-Aldrich, St. Louis, MO, USA) supplemented with 10% fetal bovine serum, 1% nonessential amino acids, 100 IU/mL penicillin, and 100 mg/mL streptomycin, and incubated at 37 °C in a humidified atmosphere of 5% CO_2_ and 95% air. After allowing the cells to attach for 24 h, they were treated with 10 μmol/L retinoic acid (Sigma-Aldrich, St. Louis, MO, USA) for 5 days to promote differentiation to a neuronal phenotype, after which the cells were 90% confluent. The medium was changed every 48 h.

### 2.2. Cell Treatments

Cells were seeded at a density of 2 × 10^6^ cells/well in 6-well plates, passaged by dissociation in 0.05% (*w*/*v*) trypsin (Sigma-Aldrich, St. Louis, MO, USA) in phosphate-buffered saline (PBS, pH 7.4) when confluent. Cells were pretreated with diosmetin (Sigma-Aldrich, St. Louis, MO, USA; Cat. # 520-34-3, purity ≥ 98%) at a different concentration (2.5, 5, 10 µmol/L) or 5 μmol/L pioglitazone (Sigma-Aldrich, St. Louis, MO, USA; Cat. # 112529-15-4, purity, ≥98%) for 1 h followed by incubation with or without various concentrations of AGEs in bovine serum albumin (Sigma-Aldrich, St. Louis, MO, USA; 50, 100, 150, 200, and 250 μg/mL) for different time points (6, 12, 24, 36, and 48 h). The concentration of diosmetin was previously protective in the cell culture model [23]. Pioglitazone, at the indicated concentration on the inhibition of AGEs, was served as the positive control group [24]. T0070907 (Sigma-Aldrich, St. Louis, MO, USA, Cat. # T8703, 5 μmol/L) was added 1 h before diosmetin or pioglitazone was stimulated [25]. Powders of diosmetin, pioglitazone, or T0070907 were dissolved in dimethyl sulfoxide (DMSO, Sigma-Aldrich, St. Louis, MO, USA) to create a 100 μmol/L stock solution, subsequently diluted in a culture medium to the appropriate concentrations for subsequent experiments. The final concentration of DMSO was less than 0.1% (*v*/*v*), which is generally innocuous to most cells [26]. The following experiments were assessed after treatment.

### 2.3. Cell Viability Assay

3-(4,5-dimethylthiazol-2-yl)-2,5-diphenyl tetrazolium bromide (MTT) assay performed by MTT Cell Proliferation Assay (cat # 30–1010K) following the manufacturer’s protocol (ATCC, Manassas, VA, USA) was used to determine the cellular viability [27]. Finally, the number of viable cells in each well was determined by a microplate reader (SpectraMax M5, Molecular Devices, Sunnyvale, CA, USA) at 570 nm. Cell viability was presented as percent viability relative to vehicle-treated control.

### 2.4. Reactive Oxygen Species Measurement

The ROS production was measured with dichloro-dihydro-fluorescein diacetate (DCFH-DA) following the manufacturer’s instructions (Sigma-Aldrich, St. Louis, MO, USA) [28]. Cells were loaded with 10 μmol/L DCFH-DA at 37 °C in a dark environment for 10 min. The fluorescence intensity of dichlorofluorescein was measured at an excitation wavelength of 488 nm and an emission wavelength of 530 nm on a microplate reader (SpectraMax M5, Molecular Devices, Sunnyvale, CA, USA). Surviving cells lysed at −80 °C, and lactate dehydrogenase (LDH) assay was carried out. DCF fluorescence readings were finally normalized to LDH absorbance to control for variations in cell number.

### 2.5. Antioxidant Enzyme Activities Determination

Determinations of SOD, GSH-Px, and CAT were performed using a SOD Activity Colorimetric Assay Kit (Cat. # K335), GSH-Px Colorimetric Assay Kit (Cat. # K762), and CAT Activity Colorimetric Assay Kit (Cat. # K773), respectively, following the procedures provided by Bio Vision, Inc. (San Francisco, CA, USA). SOD activity was assay based on the ability of SOD to inhibit nitroblue tetrazolium reduction by superoxide and following the reaction by measuring the optical density at 450 nm [29]. GSH-Px activity estimation starts with incubating the sample in the presence of hydrogen peroxide (H_2_O_2_); the decrease in nicotinamide adenine dinucleotide phosphate (NADPH) absorbance measured at 340 nm during the oxidation of NADPH to its oxidized form is indicative of GSH-Px activity [30]. CAT activity was calculated based on the decomposition rate of H_2_O_2_; the absorbance was measured at 570 nm [30]. The protein concentration was measured by using a Bio-Rad protein assay. Enzyme activities were expressed as units per milligram protein.

### 2.6. Aβ_1–40_ and Aβ_1–42_ Determination

Amyloid β (1–40) Human Assay Kit (Cat. # KHB3481) and Amyloid β (1–42) Human Assay Kit (Cat. # KHB3441) were used to quantify Aβ1–40 and Aβ1–42 in cells, respectively, based on an enzyme-linked immunosorbent assay, according to the manufacturer’s protocol (Invitrogen, Carlsbad, CA, USA). The optical density was measured at 450 nm using a microplate reader (Spectramax M5, Molecular Devices, Sunnyvale, CA, USA).

### 2.7. Caspases Activities Measurement

The caspase-12 fluorometric assay kit (Abcam plc., Cambridge, Waltham, MA, USA, Cat. # ab65664) was used to measure the caspases-12-like activity to detect substrate cleavage ATAD-AFC (ATAD: acetyl-alanine-threonine-alanine-aspartic acid; AFC: 7-amino-4-trifluoromethyl coumarin). The caspases-12-like activity was quantified by fluorescent detection of free AFC, measured at 400 nm excitation filter and 505 nm emission filter with a fluorescent microplate reader (Spectramax M5, Molecular Devices, Sunnyvale, CA, USA). The activities of caspases-9 (Cat. # ab65608) and caspases-3 (Cat. # ab39401) were determined using a caspase colorimetric assay kit, according to the manufacturer’s protocols (Abcam plc., Waltham, MA, USA). The colorimetric assay is based on the spectrophotometric detection of the chromophore p-nitroanilide (pNA) after cleavage by caspases from the labeled substrate acetyl-Leu-Glu-His-Asp-PNA (Ac-DVD-PNA, caspase-9 substrate) or acetyl-Asp-Glu-Val-Asp PNA (Ac-DVD-PNA, caspase-3 substrate), following the absorbance measured at 405 nm. All the caspase activities were relative to those obtained from the vehicle-treated control.

### 2.8. Apoptotic Cell Death Detection

The cellular DNA fragmentation ELISA kit was purchased from Roche Molecular Biochemicals (Mannheim, Germany) to qualitatively determine cytoplasmic histone-associated DNA fragments (mono- and oligonucleosomes) after induced cell death [31]. Nucleosomes were quantified by the peroxidase retained in the immunocomplex. Peroxidase was determined photometrically at 405 nm with 2,2′-azino-di[3-ethylbenzthiazolin-sulphonate] as substrate. Data are normalized to the vehicle-treated control.

### 2.9. Western Blot Analysis

Proteins (50 μg) were separated by 10% sodium dodecyl sulfate-polyacrylamide gel electrophoresis, then transferred to nitrocellulose membranes and blocked with 5% skim milk. The membranes were incubated with primary antibodies, which specifically binds to the following proteins: APP (Cell Signaling Technology, Beverly, CA, USA, Cat. # 2452), β-site APP-cleaving enzyme1 (BACE1; Santa Cruz Biotechnology, Inc., Santa Cruz, CA, USA, Cat. # sc-33711), insulin-degrading enzyme (IDE; Santa Cruz Biotechnology, Inc., Santa Cruz, CA, USA, Cat. # sc-393887), neprilysin (NEP; Santa Cruz Biotechnology, Inc., Santa Cruz, CA, USA, Cat. # sc-9149), glucose-regulated protein 78 (GRP78; Santa Cruz Biotechnology, Inc., Santa Cruz, CA, USA, Cat. # sc-13539), protein kinase R-like endoplasmic reticulum kinase (PERK; Cell Signaling Technology, Beverly, CA, USA, Cat. # 5683), phospho-PERK (Thr980) (Cell Signaling Technology, Beverly, CA, USA, Cat. # 3179), eukaryotic initiation factor 2 α (eIF2α; Cell Signaling Technology, Beverly, CA, USA, Cat. # 5324), phospho-eIF2α (Ser51) (Cell Signaling Technology, Beverly, CA, USA, Cat. # 9721), activating transcription factor 4 (ATF4; Cell Signaling Technology, Beverly, CA, USA, Cat. # 11815), C/EBP homologous protein (CHOP; Cell Signaling Technology, Beverly, CA, USA, Cat. # 2895), B-cell lymphoma 2 (Cell Signaling Technology, Beverly, CA, USA, Cat. # 3948), and Bcl-2-associated X protein (Bax; Cell Signaling Technology, Beverly, CA, USA, Cat. # 14796) overnight at 4 °C. The β-actin antibody (Santa Cruz Biotechnology, Inc., Santa Cruz, CA, USA, Cat. #sc-8432) was applied as an internal control in immunoblotting. All antibodies were utilized at a 1:1000 dilution. Blots were then washed with tris-buffered saline with 0.1% Tween^®^ 20 detergent and incubated with the secondary antibodies at room temperature for 1.5 h before visualization with chemiluminescence (Amersham Biosciences, Buckinghamshire, UK). The bands’ densities were quantified with densitometric analysis using ATTO Densitograph Software (ATTO Corp., Tokyo, Japan) and expressed as the ratio with β-actin. All values were normalized by setting the density of the control (not treated) samples as 1.0. All experimental sample values were expressed relative to this adjusted mean value.

### 2.10. Statistical Analysis

Data are expressed as the mean ± standard deviation (SD) of five independent experiments (*n* = 5) performed in triplicate. Statistically significant differences were evaluated by one-way analysis of variance and Dunnett range post hoc comparisons using Systat SigmaPlot version 14.0 (Systat Software Inc., San Jose, CA, USA). Differences were considered statistically significant at *p* < 0.05.

## 3. Results

### 3.1. Diosmetin Improved Cell *Viability*

First, we evaluate the influence of diosmetin on cell viability in AGEs-cultured SH-SY5Y cells. Cell viability was measured after incubating with AGEs in the concentration (50, 100, 150, 200, and 250 μg/mL) correlation experiment and at 6, 12, 24, 36, and 48 h in the time-course experiment. Cell viability reduced from 88.4% to 59.3% under the condition when SH-SY5Y cells were incubated with AGEs at increasing concentrations from 50 to 250 μg/mL for 24 h (Figure 1A). AGEs (200 μg/mL) yielded lower cell viability in a time-dependent tendency (Figure 1B). From the above results, SH-SY5Y cells were exposed to 200 μg/mL AGEs for 24 h to induce neuronal insults in the following experiments. Diosmetin pretreatment for 1 h before AGEs (200 μg/mL) exposure improved cell viability associated with an increasing concentration (2.5–10 μmol/L); the protective potency of diosmetin was most potent at 10 μmol/L (Figure 1C). The survival rate of AGEs (200 μg/mL)-treated SH-SY5Y cells was maintained at 85.6% when pretreated with diosmetin (10 μmol/L); similar results were obtained in AGEs (200 μg/mL)-cultured cells pretreated with 5 μmol/L of pioglitazone (cells viability, 86.4%; Figure 1C). The effect of diosmetin (10 μmol/L) or pioglitazone (5 μmol/L) on reducing AGEs-induced cell death was abolished by the PPAR*γ* antagonist T0070907 (5 μmol/L) (Figure 1D). Neither diosmetin nor pioglitazone and T0070907 alone attenuated the survival of SH-SY5Y cells (Figure 1D).

### 3.2. Diosmetin Lowered the ROS Level and Enhanced the Antioxidant Enzymes Activities

The imbalance between the excess ROS production and the impaired antioxidant system induced by the interaction between AGEs and their receptor causes irreversible neuronal damage because it increases the exposure of the neuron to oxidative stress [5]. The intracellular ROS levels in AGEs-cultured SH-SY5Y cells were 2.5-fold higher (Figure 2A). Pretreatment of SH-SY5Y cells with diosmetin (10 μmol/L) or pioglitazone (5 μmol/L) resulted in a decrease in ROS levels by 36.7% and 40.1%, respectively, in AGEs-incubated SH-SY5Y cells (Figure 2A). T0070907 pretreatment restored the actions of diosmetin and pioglitazone on the reduction in the intracellular ROS levels in AGEs-treated SH-SY5Y cells (Figure 2A). Neither diosmetin nor pioglitazone changed the ROS levels in SH-SY5Y cells without AGEs stimulation (Figure 2A).

SOD, GSH-Px, and CAT are considered the first-line defense antioxidants [32]. The activity of SOD, GSH-Px, and CAT in SH-SY5Y cells without AGEs stimulation was not significantly increased by diosmetin or pioglitazone (Figure 2B). AGEs significantly decreased the activities of SOD and GSH-Px. Pretreatment of SH-SY5Y cells with diosmetin (10 μmol/L) or pioglitazone (5 μmol/L) reversed the reductions induced by AGEs (Figure 2B). The CAT activity in AGEs-cultured SH-SY5Y cells was lower by 27.2%; pretreatment of SH-SY5Y cells with diosmetin (10 μmol/L) or pioglitazone (5 μmol/L) resulted in an increase in CAT activity to 3.0- and 3.2-fold, respectively (Figure 2B). The reversion of diosmetin or pioglitazone on AGEs induced a decrease in antioxidant activities that was not observed in T0070907 (Figure 2B).

### 3.3. Diosmetin Inhibited APP Processing and Aβ Production

It has been documented that AGEs induce ROS, thereby enhancing APP expression [5]. APP protein expression was upregulated in SH-SY5Y cells under AGEs stimulation, which was reduced when cells were pretreated with diosmetin (10 μmol/L) or pioglitazone (5 μmol/L) by 44.1 and 54.1%, respectively (Figure 3A). AGEs caused 4.1-fold and 2.9-fold increases in levels of Aβ1-40 and Aβ1-42 in SH-SY5Y cells, respectively, whereas diosmetin (10 μmol/L) or pioglitazone (5 μmol/L) pretreatment attenuated these enhancements (Figure 3B). The reduction in APP expression and Aβ production results by diosmetin or pioglitazone was blocked in the presence of 5 μmol/L T0070907 (Figure 3A,B).

The first step of the amyloidogenic pathway in processing APP by the BACE1 is to generate Aβ [33]. The higher intracellular BACE1 levels in AGEs-cultured SH-SY5Y cells were reduced by diosmetin (10 μmol/L) or pioglitazone (5 μmol/L) pretreatment with a decrease of 30.1% and 44.1%, respectively (Figure 3C). T0070907 abrogated the action of diosmetin or pioglitazone on the reduction in higher BACE1 protein expression in AGEs-cultured SH-SY5Y cells (Figure 3C).

IDE and NEP participate in amyloid degradation; deficits in both proteases may promote Aβ deposition [7]. The protein levels of IDE and NEP in AGEs-cultured SH-SY5Y cells were lowered to 43.1% and 32.7% of those in untreated controls, respectively (Figure 3D,E). The upregulation of protein levels of IDE and NEP was observed when cells received pretreatment with diosmetin (10 μmol/L) or pioglitazone (5 μmol/L), which were disappeared by 5 μmol/L T0070907 (Figure 3D,E).

### 3.4. Diosmetin Inhibited ER Stress-Activated Unfolded Protein Response

The accumulation of unfolded proteins and AGEs could disturb ER homeostasis, which is involved in the pathophysiology of AD [6]. The protein level of the ER stress sensor, GRP78, in SH-SY5Y cells under AGEs stimulation was elevated to 4.5-fold (Figure 4A). Diosmetin (10 μmol/L) or pioglitazone (5 μmol/L) pretreatment lowered the GRP78 protein level in AGEs-cultured SH-SY5Y cells to 63.5% and 55.7%, respectively (Figure 4A). The effects of diosmetin or pioglitazone were lost by 5 μmol/L T0070907 (Figure 4A).

A body of evidence has demonstrated that the pathogenesis and progression of neurodegenerative diseases, including AD, are closely associated with the ER stress-mediated PERK signaling pathway involving eIF2α phosphorylation, thereby activatingATF4,which can determine the upregulation of the proapoptotic factor CHOP [34]. The ratios of phosphoprotein to total protein in PERK (p-PERK/PERK) and eIF2α (p-eIF2α/eIF2α) were about 3.8-fold more significant in the AGEs-cultured SH-SY5Y cells (Figure 4B,C). The elevation ratios of p-PERK/PERK and p-eIF2α/eIF2α induced by AGEs decreased by pretreatment with pioglitazone (5 μmol/L) to 2.1–2.2-fold, respectively (Figure 4B,C). T0070907 (5 μmol/L) suppressed the effects of diosmetin or pioglitazone. Neither diosmetin nor pioglitazone and T0070907 changed the protein levels of PERK and eIF2α in AGEs-cultured SH-SY5Y cells (Figure 4B,C).

AGEs-induced upregulation on ATF4 and CHOP protein expression in SH-SY5Y cells was remarkably reversed in diosmetin (10 μmol/L) pretreatment (27.7% decrease in ATF4, 38.1% decrease in CHOP, relative to those in their vehicle-treated counterparts; Figure 4D,E). The protein levels of ATF4 and CHOP were decreased to 65.3 and 61.8%, respectively, in AGEs-cultured SH-SY5Y cells pretreated with 5 μmol/L pioglitazone relative to those of their vehicle-treated counterparts (Figure 4D,E). T0070907 (5 μmol/L) abolished the effects of diosmetin or pioglitazone on the revising higher ATF4 and CHOP protein expression in AGEs-cultured SH-SY5Y cells (Figure 4D,E).

### 3.5. Diosmetin Suppressed ER Stress-Mediated Apoptosis

CHOP involved in the ER stress-induced apoptotic pathway could regulate the expression of the Bcl-2 protein family [35]. AGEs elevated the Bax/Bcl-2 ratio in SH-SY5Y cells compared with the vehicle-treated group, which was attenuated by pretreatment with diosmetin (10 μmol/L) or pioglitazone (5 μmol/L); T0070907 (5 μmol/L) abrogated the effects of diosmetin or pioglitazone (Figure 5A).

Events mediate apoptosis, including caspases activation [36]. ER stress-specific caspase cascade comprises caspase-12, -9, and -3, by the order [36]. There was a 3.7-fold increase in caspase-12-like activity in AGEs-cultured SH-SY5Y cells, which was reduced by 48.3 and 52.9%, respectively, when cells were pretreated with diosmetin (10 μmol/L) or pioglitazone (5 μmol/L; Figure 5B). In the presence of diosmetin (10 μmol/L), the increased activities of caspase-9 and caspase-3 in SH-SY5Y cells exposed to AGEs were downregulated by 56.7% and 47.6%, respectively, to those observed on the vehicle-treated counterparts (Figure 5B). SH-SY5Y cells pretreated with pioglitazone (5 μmol/L) with higher caspase-9 and caspase-3-like activities caused by AGEs were lowered to 51.2 and 41.4%, respectively, to those in their vehicle-treated counterparts (Figure 5B). The effects of diosmetin (10 μmol/L) and pioglitazone (5 μmol/L) on lowering the caspases activities in AGEs-incubated SH-SY5Y cells were disappeared in the presence of T0070907 (5 μmol/L).

DNA fragmentation is a unique feature of apoptosis [31]. The extent of apoptotic DNA fragmentation in SH-SY5Y cells under AGEs stimulation decreased by 50.3% in pretreatment with diosmetin (10 μmol/L, Figure 5C); similar results were obtained in the pioglitazone (5 μmol/L) pretreated group. The effects of diosmetin or pioglitazone on the elimination of higher apoptotic DNA fragmentation in AGEs-cultured SH-SY5Y cells were not shown in the 5 μmol/L of T0070907 pretreatment (Figure 5C).

## 4. Discussion

Polyphenolic compounds, such as flavonoids, are commonly produced as secondary metabolites in plants of citrus species [15]. Diosmin is clinically applied in vascular system disorders among the natural flavonoids, especially in chronic venous insufficiency [16]. However, the amount of diosmetin detected in plasma after a single oral administration of diosmin is low, highly variable, and often inconsistent across different studies [19]. The micronization process could now be considered a consolidated technology to improve drug dissolution and intestinal permeability, but further increases in oral bioavailability are required to achieve optimal therapeutic efficacy [19]. Although diosmin reducing Aβ-associated pathology in AD mouse models has been documented [17], the protective mechanisms were related to ameliorating oxidative stress, and Aβ-mediated neurotoxicity under glycation is essential to be elucidated. Considering the biological effects of oral diosmin may be mediated by its aglycone diosmetin [16], we thus used AGEs-induced SH-SY5Y cells as an in vitro Alzheimer-like model to clarify the protective effects of diosmetin.

ROS production induced by the interaction between AGEs and their receptors increases the exposure of the neuron to oxidative stress and induces irreversible neuronal damage [6]. Diosmetin has been reported to intensify the cellular antioxidant system by preventing intracellular ROS generation and increasing the intracellular antioxidant enzymes activities [23]. Pretreatment of SH-SY5Y cells with diosmetin at a concentration of 10 μmol/L was capable of reducing AGEs-induced oxidative stress by upregulating the antioxidant enzyme activities of SOD, GPx, and CAT. However, diosmetin alone did not change the ROS levels and the activity of SOD, GSH-Px, and CAT in SH-SY5Y cells without AGEs stimulation. It appears that diosmetin attenuated oxidative stress induced by AGEs via enhancing the expression of antioxidant enzymes and was able to lead to ROS scavenging. Our data support the hypothesis that diosmetin counteracts AGEs-triggered oxidative damage in SH-SY5Y cells, contributing to a balance between oxidative stress and antioxidant defense systems.

AGEs-induced ROS increase Aβ synthesis through the proteolytic processing of a transmembrane protein, APP, by upregulating APP processing protein, such as BACE1 [33]. The Aβ levels depend not only on Aβ production but also on impairment of Aβ degradation or clearance, which may play a key role in AD pathogenesis [7]. Several enzymes are described to cleave Aβ, among which NEP and IDE are the most essential and promising Aβ-degrading candidates [7]. Aβ has a length of 39–43 amino acid residues; the predominant Aβ isoforms are 40 amino acid residues (Aβ1-40) and 42 amino acid residues (Aβ1-42); the latter has a higher propensity to form prefibrillar aggregates [37]. Diosmetin pretreatment resulted in translational downregulation of APP and BACE1 in parallel with upregulation of IDE and NEP protein levels, thus resulting in a decrease in Aβ secretion. Therefore, these findings support the notion that downregulation of Aβ generation and promoting Aβ degradation are the responsible mechanisms by which diosmetin protects against AGEs-induced neuronal cellular damage.

The accumulation of AGEs is associated with ER stress induction in cells, contributing to the etiology and pathogenesis of multiple human diseases, including AD [6]. Cells have evolved various protective strategies to combat the deleterious effects of ER stress [6]. ER stress uncouples the interaction between GRP78 and ER stress sensors, which activates the unfolded protein response (UPR) branches, leading to restoring ER homeostasis. In contrast, the PERK/eIF2α/ATF4 axis activation is sustained under prolonged ER stress conditions, thus activating the transcription of genes involved in cell apoptosis CHOP protein [38]. Multiple evidence demonstrated that the pathogenesis and progression are closely associated with the PERK-dependent UPR signaling pathway. Therapeutic targeting of the PERK-dependent UPR signaling branches could be a ground-breaking treatment for slowing down AD [34]. We observed that diosmetin pretreatment countered AGEs-induced upregulation of ER stress-related proteins involving GRP78, p-PERK, p-eIF2α, and ATF4 in SH-SY5Y cells. Otherwise, diosmetin pretreatment attenuated the increasing CHOP expression level in SH-SY5Y cells after induction with AGEs. These data suggested that diosmetin contributing to the neuron protection in AGEs-treated SH-SY5Y cells was mediated by suppressing ER stress-induced GRP78 to downregulate the PERK/eIF2α/ATF4/CHOP pathway.

Although Bcl-2 family members are thought to function principally at the mitochondrial outer membrane, there is ample evidence that they influence homeostasis and apoptosis from the ER [38]. As a transcriptional factor, CHOP represses the Bcl-2 gene expression resulting in downregulating the anti-apoptotic Bcl-2 protein, but it has been shown to promote pro-apoptotic proteins, such as Bax, in the cell [38]. We observed that diosmetin prevented AGEs-induced Bcl-2 protein downregulation and, in parallel, lowered Bax levels in SH-SY5Y cells. Diosmetin-induced reductions in the AGE-induced ER stress inhibited neuronal apoptosis by restoring the balance between the pro-and anti-apoptotic members of the Bcl-2 family, which could be considerable.

Caspase-dependent neuronal death contributes to neuron loss, and thus, caspase inhibition offers some hope for extending AD neuron survival, so that other agents targeting upstream events may delay or reverse primary AD pathology [36]. Prolonged ER stress can activate caspase-12, which may feed back into a caspase-9-dependent loop; caspase-9 activates effector caspases, such as caspase-3, resulting in a cleavage of cellular proteins and cell demise by apoptosis [36]. Caspase-mediated proteolysis cleaves protein substrates into fragments [31]. Compared with AGEs-treated SH-SY5Y cells, diosmetin-pretreated cells had a minor apoptotic DNA fragmentation and decreased cleavage of caspase-12, -9, -3. Our data suggest that diosmetin prevents the AGEs-induced activation of apoptosis in SH-SY5Y cells by reducing the caspase 12-dependent ER stress. The blockage of ER stress-mediated neuronal apoptosis by diosmetin may be a potential target for AGEs-triggered aggravation of AD pathology.

PPAR*γ* is expected to provide a new therapeutic approach for the treatment of atherosclerotic and ischemic cerebrovascular diseases, and it is essential for the prevention of neurodegenerative diseases, including AD [39]. PPAR*γ* protection of the brain from ischemic cerebrovascular diseases by suppression of ER stress-related CHOP protein has been documented [40]. PPAR*γ* agonists may also attenuate ER stress induced by saturated fatty acids through the downregulation of the PERK/CHOP signaling pathway, contributing to the anti-atherosclerotic effects of pioglitazone [41]. There is a similar effect between diosmetin and positive pioglitazone control against Aβ-induced neurotoxicity and ER stress-mediated neuronal apoptosis caused by AGEs, revealing that the action of diosmetin might be mediated by PPAR*γ* activation. T0070907 is a potent and selective irreversible PPAR*γ* antagonist with a half-maximal inhibitory concentration equal to 1 nmol/L, which displays 800-fold selectivity for PPAR*γ* over PPARα and PPARδ [25]. The suppression of diosmetin on AGEs-mediated induction of Aβ accumulation and neuronal ER stress was abated by T0070907, strongly indicating that the protective effects against AGEs-induced injury in neuronal cells by diosmetin might be dependent on PPAR*γ* activation. These data suggested that PPAR*γ* might be a potential target of diosmetin, acting to alleviate ER stress and cell death caused by Aβ protein under glycation in AD.

It has been documented that treating mice with diosmetin can improve the impaired memory and cognition induced by chronic stress by increasing the antioxidant capacity of brain tissue and serum and improving serum corticosterone levels [42]. Although diosmetin is considered permeable across the BBB, lactoferrin modified long-circulating liposomes for brain-targeted delivery of diosmetin have been developed with potential implications for AD disease treatment [43]. However, further in vivo studies or clinical investigations are necessary to estimate the compound’s toxicity, efficacy, and dosage regimens to identify diosmetin’s advantages and disadvantages in drug development.

## 5. Conclusions

In conclusion, diosmetin protects SH-SY5Y cells against AGEs-induced ROS, decreasing the Aβ accumulation via downregulation of Aβ generation and enhancing Aβ degradation. In addition, diosmetin resolved ER stress-mediated PERK/eIF2α/ATF4/CHOP activation and anti-apoptotic effects by suppressing the caspase 12-dependent pathway in AGEs-treated SH-SY5Y cells. The improvements of diosmetin on AGEs-induced deficits in SH-SY5Y cells might be through the activation of the PPAR*γ* pathway. This investigation builds a foundation for cell-based confirmed diosmetin as a promising candidate for combating AGEs-related AD. Its practical therapeutic effects on AD require further research.

## Figures and Tables

**Figure 1 nutrients-14-02248-f001:**
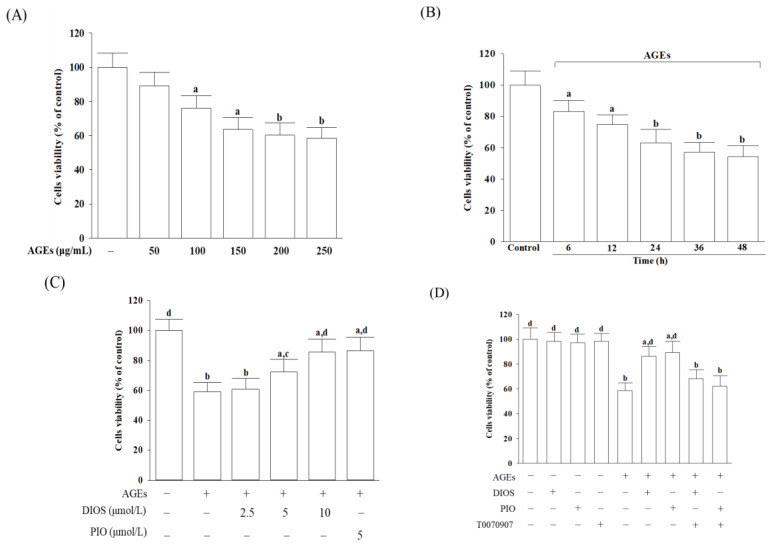
Effects on the viability of AGE-cultured SH-SY5Y cells. (**A**) Various concentrations of AGEs were incubated with SH-SY5Y cells for 24 h. (**B**) SH-SY5Y cells were incubated with 200 μg/mL AGEs for various lengths of time. (**C**) SH-SY5Y cells were pretreated with various concentrations of diosmetin (DIOS; 2.5–10 µmol/L) or pioglitazone (PIO; 5 μmol/L) for 1 h, then exposed to 200 μg/mL AGEs for another 24 h. (**D**) T0070907 (5 μmol/L) was added 1 h before diosmetin (10 µmol/L) or pioglitazone (5 μmol/L) were stimulated. Cell viability was determined using an MTT assay and expressed as a percentage of the untreated cells, considering the control group. The results are shown as the mean ± SD of five independent experiments (*n* = 5), each performed in triplicate. ^a^ *p* < 0.05 and ^b^ *p* < 0.01 compared to the data from untreated control group (control). ^c^ *p* < 0.05 and ^d^ *p* < 0.01 compared to the data from cells cultured under AGEs without any treatment.

**Figure 2 nutrients-14-02248-f002:**
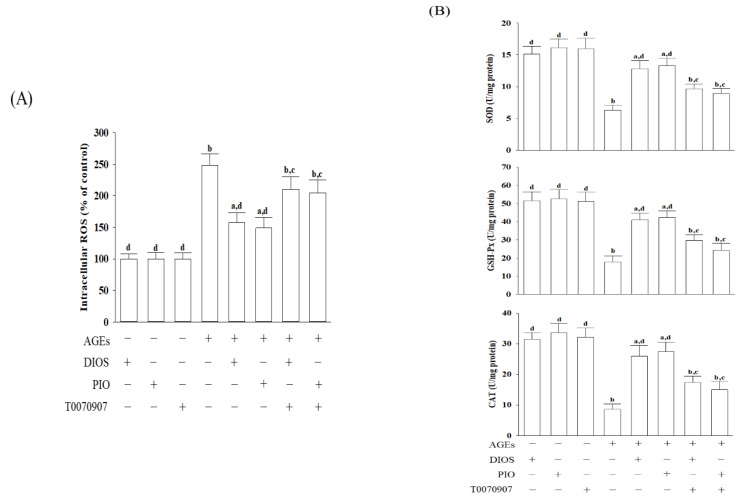
Effects on the level of ROS and the activities of antioxidant enzymes in AGE-cultured SH-SY5Y cells. Cells were pretreated with diosmetin (DIOS; 10 µmol/L) or pioglitazone (PIO; 5 μmol/L) for 1 h, then incubated with or without 200 μg/mL AGEs for another 24 h. T0070907 (5 μmol/L) was added 1 h before diosmetin or pioglitazone were stimulated. (**A**) The intracellular level of ROS was measured using the oxidation-sensitive fluoroprobe DCFH-DA. (**B**) Qualified commercial assay kits were used to determine the activities of SOD, GSH-Px, and CAT. The results are presented as the mean ± SD of five independent experiments (*n* = 5), each performed in triplicate. ^a^ *p* < 0.05 and ^b^ *p* < 0.01 compared to the data from untreated control group (control). ^c^ *p* < 0.05 and ^d^ *p* < 0.01 compared to the data from cells cultured under AGEs without any treatment.

**Figure 3 nutrients-14-02248-f003:**
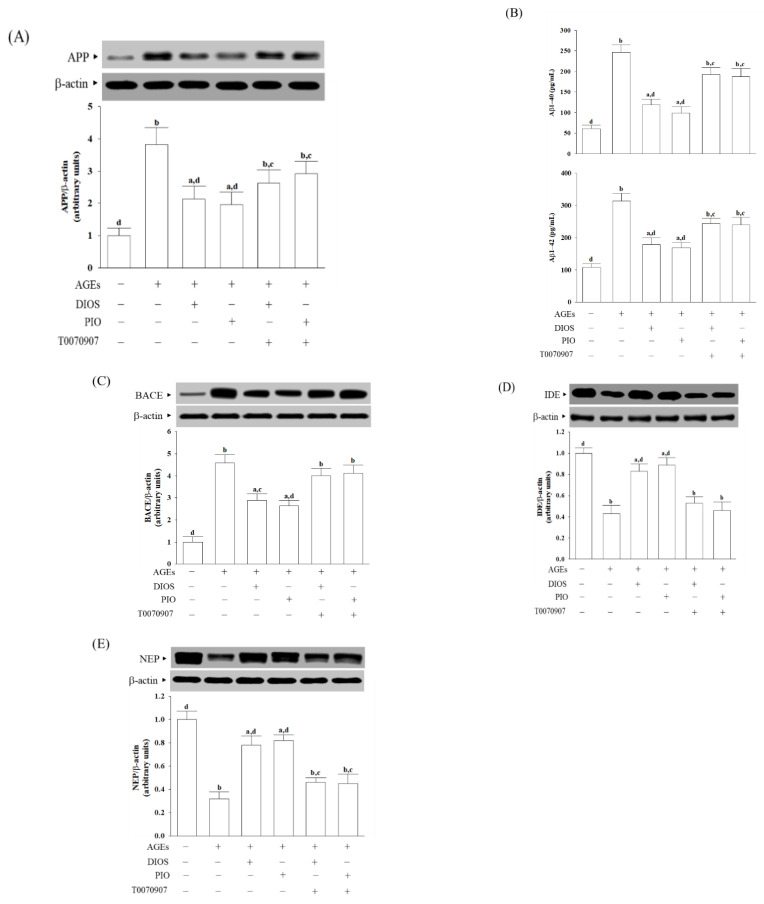
Effects on APP processing and Aβ production in AGEs-treated SH-SY5Y cells. Cells were pretreated with diosmetin (DIOS; 10 µmol/L) or pioglitazone (PIO; 5 μmol/L) for 1 h, then exposed to 200 μg/mL AGEs for another 24 h. T0070907 (5 μmol/L) was added 1 h before diosmetin or pioglitazone were stimulated. (**A**) A representative Western blot analyzing the relative levels of APP. (**B**) The levels of Aβ1-40 and Aβ1-42 were determined by ELISA assay. The protein levels of (**C**) BACE, (**D**) IDE, and (**E**) NEP were also detectable by Western blot. Densitometric values of specific protein bands were normalized to β-actin intensity. The results are shown as the mean ± SD of five independent experiments (*n* = 5), each performed in triplicate. ^a^ *p* < 0.05 and ^b^ *p* < 0.01 compared to the data from untreated control group (control). ^c^ *p* < 0.05 and ^d^ *p* < 0.01 compared to the data from cells cultured under AGEs without any treatment.

**Figure 4 nutrients-14-02248-f004:**
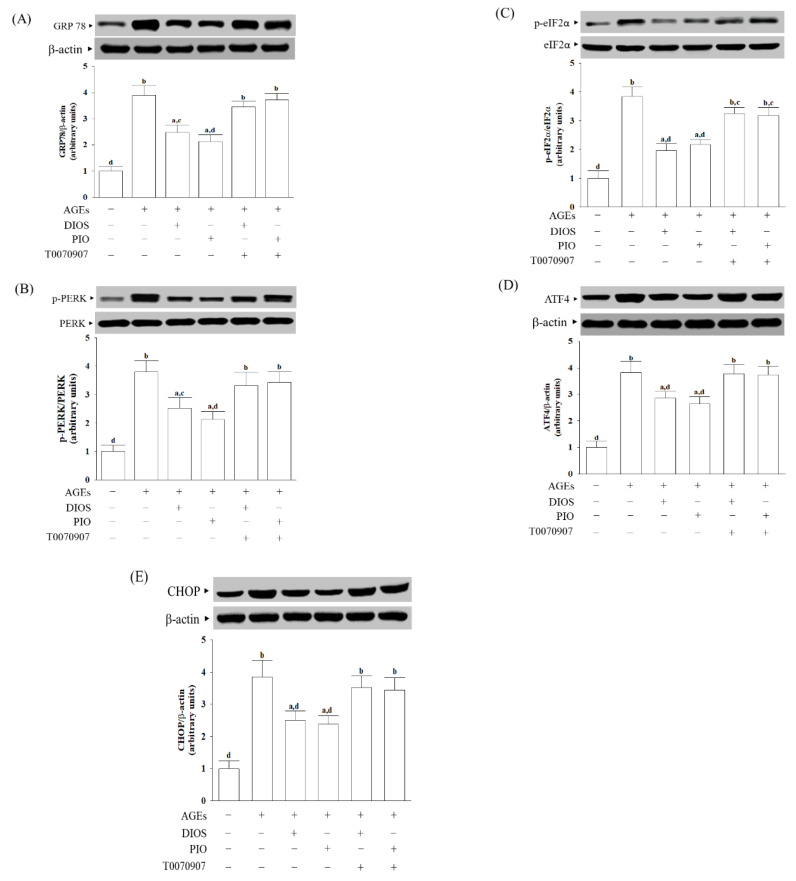
Effects on the ER stress-activated UPR in AGE-cultured SH-SY5Y cells. Cells were pretreated with diosmetin (DIOS; 10 µmol/L) or pioglitazone (PIO; 10 μmol/L), then exposed to 200 μg/mL AGEs for another 24 h. T0070907 (5 μmol/L) was added 1 h before diosmetin or pioglitazone were stimulated. Western blot was carried out to detect the expression of (**A**) GRP78, (**B**) PERK, (**C**) eIF2α, (**D**) ATF4, and (**E**) CHOP. The band densities for GRP78, ATF4, and CHOP were normalized to β-actin band intensity. The ratios of phosphoprotein to total protein in PERK (p-PERK/PERK) and eIF2α (p-eIF2α/eIF2α) were calculated. The results are shown as the mean ± SD of five independent experiments (*n* = 5), each of which was performed in triplicate. ^a^ *p* < 0.05 and ^b^ *p* < 0.01 compared to the data from untreated control group (control). ^c^ *p* < 0.05 and ^d^ *p* < 0.01 compared to the data from cells cultured under AGEs without any treatment.

**Figure 5 nutrients-14-02248-f005:**
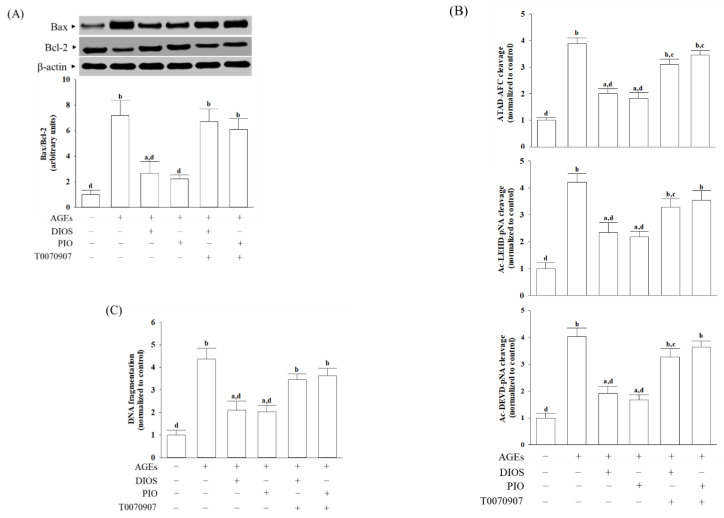
Effects on ER stress-associated apoptosis in AGEs-treated SH-SY5Y cells. Cells were pretreated with diosmetin (DIOS; 10 µmol/L) or pioglitazone (PIO; 5 μmol/L) for 1 h, then exposed to 200 μg/mL AGEs for another 24 h. T0070907 (5 μmol/L) was added 1 h before diosmetin or pioglitazone were stimulated. (**A**) Representative photographs of Western blot analysis for Bax and Bcl-2. The ratio of the relative intensities of Bax to Bcl-2 (Bax/Bcl-2) was shown. (**B**) Analysis of caspase-12,-9-, and -3-like activity. The caspases-12-like activity in cell lysates was quantified by fluorescent detection of cleavage of substrate ATAD-AFC. Caspase-9 and caspase-3-like activities were measured against the colorimetric substrates Ac-LEHD-pNA and Ac-DEVD-pNA, respectively. (**C**) Apoptosis was determined by measuring cytoplasmic histone-associated DNA fragments using the cell death detection ELISAplus kit. The results are shown as the mean ± SD of five independent experiments (*n* = 5), each performed in triplicate. ^a^ *p* < 0.05 and ^b^ *p* < 0.01 compared to the data from untreated control group (control). ^c^ *p* < 0.05 and ^d^ *p* < 0.01 compared to the data from cells cultured under AGEs without any treatment.

## Data Availability

All the data needed to evaluate the conclusions in the paper are present in the paper. Additional data related to this paper may be requested from the authors.

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
