# Peer review of "Diosmetin Targeted at Peroxisome Proliferator-Activated Receptor Gamma Alleviates Advanced Glycation End Products Induced Neuronal Injury"

_nutrients, 2022, doi:10.3390/nu14112248_

Round 1
Reviewer 1 Report
Manuscript titled as “Diosmetin Targeted at Peroxisome Proliferator-Activated Receptor Gamma Alleviates Advanced Glycation End Products Induced Neuronal Injury” has a significant pros and cons for the special issue for “Impact of Nutrition or FDA-Approved Medicine Repurposing Utilization on Metabolic Syndrome and Diabetic Complications”. Nutrients-1721004 suggested the potential repositioning potency of diosmetin in AGE-inducible neural disorder. However, in nutrients-1721004 authors solely relied on the in vitro results and thus, it is not convincing whether diosmetin may be delivered through BBB successfully in mammals.
[1] Diosmetin has a poor solubility (Sun et al. Journal of Functional Foods 2022); therefore, diosmetin itself may delivered as the experimental design in nutrients-1721004. Authors should acknowledge the limitation of the delivery method of diometin.
[2] Also, authors did not provide in vivo data; therefore, authors should describe more detailed information to prove the practical aspect since the special issue focused on the repositioning of the approved chemical.
[3] Unnecessary italic has found numerous parts in nutrients-1721004. (i.e. L17-18, 28, 38, 93, 346, 432, and so on)
[4] In figure 1, there is no statistical values listed on AGE – (A/B).
[5] [L243] ROS excess – excess ROS
[6] [L302] Western is not a human name; therefore, small letter should be used.
[7] [L318] Spacing issue.
[8] Recommend to provide visual pictures for Fig 5C.
Author Response
Dear distinguished referee:
Thank you very much for reading this manuscript and for the helpful comments. The revision has been amended according to your kind suggestions as follows.
[1] Diosmetin has a poor solubility (Sun et al. Journal of Functional Foods 2022); therefore, diosmetin itself may delivered as the experimental design in nutrients-1721004. Authors should acknowledge the limitation of the delivery method of diometin.
The limitation of the delivery method of diometin has been described in line 427-431 according to your recommendation. We hope this improvement will be satisfactory and acceptable.
[2] Also, authors did not provide in vivo data; therefore, authors should describe more detailed information to prove the practical aspect since the special issue focused on the repositioning of the approved chemical.
Although we did not provide in vivo data, it has been documented that treating mice with diosmetin can improve the impaired memory and cognition induced by chronic stress by increasing the antioxidant capacity of brain tissue and serum and improving serum corticosterone level (Saghaei et al., Evid. Based Complement. Alternat. Med. 2020, 2020). Diosmetin is permeable across the BBB and has been considerable developed with potential implications in AD disease treatment. Please find it in line 519-527. We hope this improvement will be satisfactory and acceptable.
[3] Unnecessary italic has found numerous parts in nutrients-1721004. (i.e. L17-18, 28, 38, 93, 346, 432, and so on)
The unnecessary italic style has been corrected throughout the revision according to your recommendation. We hope this improvement will be satisfactory and acceptable.
[4] In figure 1, there is no statistical values listed on AGE – (A/B).
Thank you for your recommendations; the statistical values are shown in figure 1.
[5] [L243] ROS excess – excess ROS
The indicated sentence (line 255) has been corrected according to your recommendation. Thank you very much.
[6] [L302] Western is not a human name; therefore, small letter should be used.
According to your recommendation, the word “Western” has been changed to a small letter (line 338). Thank you very much.
[7] [L318] Spacing issue.
According to your recommendation, the spacing issue has been solved in line 352. Thank you very much.
[8] Recommend to provide visual pictures for Fig 5C.
The apoptotic response was measured by the cell death detection ELISA plus kit. Data comparing vehicle-treated cells in the absence of test substance was commonly acceptable. We hope this illustration will be satisfactory and acceptable.
According to the reviewers’ comments, the changes in the revision are highlighted in red. We hope that this revised version of our work will meet your high standards for acceptance. Also, I wish to express the warmest thanks to you again. Your kind agreement of recognition will be sincerely appreciated.
Reviewer 2 Report
The paper is interesting and the results described and commented extensively. In the conclusions I would not use the word "drug" to define diosmetine but rather the word "compound" or other alternative.
Author Response
Dear distinguished referee:
Thank you very much for reading this manuscript and for the helpful comments. The revision has been amended according to your kind suggestions as follows.
The paper is interesting and the results described and commented extensively. In the conclusions I would not use the word "drug" to define diosmetin but rather the word "compound" or other alternative.
The word "compound" (line 534) has been deleted according to your recommendation. Thank you very much.
According to the reviewers’ comments, the changes in the revision are highlighted in red. We hope that this revised version of our work will meet your high standards for acceptance. Also, I wish to express the warmest thanks to you again. Your kind agreement of recognition will be sincerely appreciated.